# Cloud Feedbacks from CanESM2 to CanESM5.0 and their Influence on Climate Sensitivity

John, G. Virgin[1], Christopher, G. Fletcher[1], Jason, N. S. Cole[2], Knut von Salzen[2], and Toni Mitovski[2,3]

[1]Department of Geography & Environmental Management, University of Waterloo, Waterloo, Ontario, Canada
[2]Canadian Centre for Climate Modelling and Analysis, Environment and Climate Change Canada, Victoria, British Columbia, Canada
[3]Ministry of Health, Government of British Columbia, Victoria, British Columbia, Canada

**Correspondence:** John G. Virgin (jgvirgin@uwaterloo.ca)

**Abstract.**

The newest iteration of the Canadian Earth System Model (CanESM5.0.3) has an Effective Climate Sensitivity (ECS) of 5.65 kelvin, which is a 54% increase relative to the model's previous version (CanESM2 - 3.67 K), and the highest sensitivity of all current models participating in the sixth phase of the coupled model inter-comparison project (CMIP6). Here, we explore the underlying causes behind CanESM5's increased ECS via comparison of forcing and feedbacks between CanESM2 and CanESM5. We find only modest differences in radiative forcing as a response to $CO_2$ between model versions. We find small increases in the surface albedo and longwave cloud feedback, as well as a substantial increase in the SW cloud feedback in CanESM5. Through the use of cloud area fraction output and cloud radiative kernels, we find that more positive low and non-low shortwave cloud feedbacks— particularly with regards to low clouds across the equatorial Pacific, as well as sub/extratropical free troposphere cloud optical depth— are the dominant contributors to CanESM5's increased climate sensitivity. Additional simulations with prescribed sea surface temperatures reveal that the spatial pattern of surface temperature change exerts controls on the magnitude and spatial distribution of low cloud fraction response, but does not fully explain the increased ECS in CanESM5. The results from CanESM5 are consistent with increased ECS in several other CMIP6 models, which has been primarily attributed to changes in shortwave cloud feedbacks.

## 1 Introduction

Equilibrium Climate Sensitivity (ECS), defined as the global, annual mean surface warming the Earth would exhibit as a response to a doubling of $CO_2$, is a frequently cited emergent property from simplified climate models (Charney et al., 1979), as well as modern Earth system models (Andrews et al., 2012; Vial et al., 2013). The first estimates of ECS from Earth system models ranged from 1.5 - 4.5 K (Charney et al., 1979). In the latest phase of the Coupled Model Inter-comparison Project (CMIP6), the range of ECS from participating models has widened (1.8 - 5.5 K), with the mean shifting towards higher values

than the previous phase of CMIP (3.2 to 3.7 K from CMIP5 to CMIP6) (Flynn and Mauritsen, 2020; Zelinka et al., 2020). Inter-model spread of ECS is primarily attributed to radiative feedbacks on the climate system- specifically with regards to cloud feedbacks, which are the primary source of spread across models (Caldwell et al., 2016; Vial et al., 2013; Dufresne and

Bony, 2008).

Understanding cloud feedback uncertainty and its influence on the ECS of Earth system models (ESMs) has been an imperative of researchers in recent decades- particularly with regards to properties such as cloud optical depth, which determine the amount of reflected shortwave (SW) radiation and thus help cool the planet (Vial et al., 2013; Tan et al., 2016; Zelinka et al., 2020; Bjordal et al., 2020). SW cloud feedbacks can be separated based on latitude; middle latitude SW cloud feedbacks are

mostly negative from the optical thickening of clouds due to phase transition towards liquid in ice/mixed phase clouds (Goosse et al., 2018; Senior and Ingram, 1989). In high latitudes, sea ice loss exposes the ocean surface and increases surface turbulent fluxes, and therefore humidity, which increases low level cloudiness (Goosse et al., 2018). In low latitudes, the SW cloud feedback is robustly positive in both ESMs and Large Eddy Simulations (LES), owing to a reduction in the fraction and thickness of marine shallow cumulus and stratocumulus clouds near the planetary boundary layer (PBL) (Bretherton and Blossey, 2014;

Bretherton et al., 2013; Ceppi et al., 2017). The physical mechanisms behind SW low cloud feedbacks are tied to multiple thermodynamic, radiative, and dynamical processes- termed cloud controlling factors (CCFs) (Klein et al., 2017). Specifically, mechanisms favoring an increase in low cloud fraction in baseline climatology regimes include stronger PBL temperature inversions (Wood and Bretherton, 2006; Klein and Hartmann, 1993; Bretherton et al., 2013), colder sea surface temperatures (SST) (Bretherton and Blossey, 2014), less subsidence (Blossey et al., 2013), and increased free troposphere humidity (Van der

Dussen et al., 2015).

While the sensitivity of marine low cloud cover (LCC) to specific factors varies significantly from model to model, differing sensitivities to SSTs have been identified as a explanatory factor for spread across ESMs (Qu et al., 2014). This link suggests the spatial pattern of surface warming has important implications for low cloud responses (Rose et al., 2014; Zhou et al., 2015), and therefore the SW cloud feedback and climate sensitivity (Andrews and Webb, 2018).

Here, we investigate the causes of increased climate sensitivity in the newest version of the Canadian Earth System Model, which is the highest of all models currently participating in CMIP6 (Flynn and Mauritsen, 2020). We examine CanESM5's high ECS particularly in relation to the previous model version that was contributed to CMIP5 (CanESM2). With a particular focus on decomposed cloud feedbacks, we quantify the differences in both forcing and feedback between CanESM2 and CanESM5 in order to establish a physical link for the shift in ECS. Lastly, we examine the spatial pattern of warming in CanESM5 and

its influence of both global mean and local cloud feedbacks as a potential explanatory variable for CanESM5's high ECS.

## 2    Methods

### 2.1    Models

We compare two versions of CanESM in this study. CanESM2, the second generation earth system model from the Canadian Centre for Climate Modelling and Analysis (CCCma), consists of their fourth generation atmosphere model (CanAM4), land

surface model (CLASS), terrestrial carbon model (CTEM), CSM ocean model from the National Centre for Atmosphere Research (NCAR), and ocean carbon model (CMOC) (von Salzen et al., 2013; Arora et al., 2009; Zahariev et al., 2008; Arora et al., 2011; Gent et al., 1998). CanESM5 (Swart et al., 2019), is the newest generation of the Canadian Earth System Model, uses an updated version of CLASS (version 2.7 to 3.6.2) (Verseghy, 2000), CanNEMO for the ocean model, which is based on NEMO3.4.1 (Madec, 2012), and Louvain-la-Neuve sea ice model (LIM2) (Fichefet and Maqueda, 1997; Bouillon et al., 2009).

The fifth generation atmospheric model (CanAM5) has the same horizontal resolution as CanAM4 while increasing the vertical layers from 35 to 49 with majority of the additional layers added to the upper troposphere and lower stratosphere. While there are a number of improvements to radiative transfer, aerosol, and surface parameterization, changes to cloud parameterizations are discussed briefly given their direct potential connection to cloud feedbacks. Ice cloud parameterizations in CanAM5 largely remain as in CanAM4 (von Salzen et al., 2013) with the exception of adjustments to uncertain parameters. For liquid clouds, the primary change is autoconversion of cloud liquid to rain in CanAM5, which now uses the parameterization of Wood (2005) instead of the parameterization of Khairoutdinov and Kogan (2000) which was used in CanAM4. The change in autoconversion parameterization includes the second indirect aerosol effect— a process not considered in CanAM4 (von Salzen et al., 2013).

## 2.2 Forcing-Feedback Analysis

We consider energy balance at Earth's top of atmosphere (TOA) using the following equation:

$$N = F - \lambda \Delta T_s \tag{1}$$

Where $N$ is net radiation imbalance (in $Wm^{-2}$), $F$ is the Effective Radiative Forcing (ERF) due to that of an external agent (e.g. $CO_2$, in $Wm^{-2}$), $\Delta T_s$ is the global, annual mean surface temperature response (in Kelvin), and $\lambda$ is the net Climate Feedback Parameter (in $Wm^{-2}K^{-1}$). Equation 1 assumes a linear relationship between radiation imbalance and surface temperature response (i.e. a constant $\lambda$). Under this assumption, an earth system model with a stronger (more negative) $\lambda$ term will reestablish energy balance faster- and with a weaker surface temperature response- than a weak $\lambda$ term. We calculate the net feedback parameter using pre-industrial control and abrupt-4xCO2 experiments for each version of CanESM. For CanESM2, we use 150 years of pre-industrial control and abrupt 4xCO2 coupled model output submitted to to the Earth System Grid Federation under run 1, initialization 1, and physics 1 (r1i1p1) (Taylor et al., 2012). For CanESM5, we use the same experiments submitted for the core CMIP6 experiment deck (Eyring et al., 2016).

The surface temperature response after the system has reached equilibrium ($N = 0$ $Wm^{-2}$) is defined as the Equilibrium Climate Sensitivity, which is typically measured under a 2xCO2 ERF ($ECS = -F/\lambda$). We use the term "Effective" above, as opposed to "Equilibrium" Climate Sensitivity, given the linear assumptions in Equation 1, where Equilibrium Climate Sensitivity is representative of warming once the system as reached true equilibrium and does not require any form of statistical extrapolation (Knutti et al., 2017). The extent that the linear approximation accurately represents both forcing and feedback varies from model to model, where some models exhibit a more linear response to $CO_2$ than others (Andrews et al., 2012). A

time varying net feedback parameter has been- at least partially- attributable to differences in model timescale and magnitude of "rapid adjustments" in the climate system, where quick tropospheric climate response to $CO_2$ modifies TOA energy balance (Smith et al., 2018; Forster et al., 2013; Sherwood et al., 2015).

We consider the influence of rapid adjustments by diagnosing the ERF using two distinct methods. First, we use an ordinary least squares linear regression between TOA radiation imbalance ($N$) and surface temperature response ($\Delta T_s$) in abrupt 4xCO$_2$ experiments, where the extrapolated y-intercept of the regression line equals 2x the ERF (ERF$_g$, Gregory et al. (2004)). Second, we use 30 year fixed sea surface temperature experiments (piClim-control and piClim-4xCO$_2$) submitted to the Radiative Forcing Model Inter-comparison Project (RFMIP) (Pincus et al., 2016). The ERF is calculated by differencing net TOA radiation in

30 year annual mean fixed sea surface temperature experiments (ERF$_h$), where one experiment uses pre-industrial control $CO_2$ and the other uses abrupt 4xCO$_2$ (Hansen et al., 2005; Pincus et al., 2016). Under the ERF$_h$ definition, both tropospheric and stratospheric rapid adjustments from clouds, air temperature, water vapour, and surface albedo are included along with $CO_2$.

Using the Gregory regression method, we obtain the net feedback parameter as the slope of the regression line. Furthermore, we quantify the ECS for an abrupt 2xCO$_2$ forcing as the extrapolated x-intercept of the regression line divided by two. We

consider the net feedback parameter as the linear sum of individual radiative feedbacks within the climate system:

$$\lambda = \lambda_p + \lambda_{lr} + \lambda_{wv} + \lambda_a + \lambda_c + R_e \tag{2}$$

Where the net feedback parameter is made up of contributions from the Planck ($\lambda_p$), lapse rate ($\lambda_{lr}$), water vapour ($\lambda_{wv}$), surface albedo ($\lambda_a$), and cloud ($\lambda_c$) feedbacks. A residual term is also included ($R_e$) in order to account for nonlinearities. We use a combination of the radiative kernel and Gregory regression methods to diagnose individual radiative feedbacks

(Block and Mauritsen, 2013; Soden and Held, 2006). Specifically, we use six sets of radiative kernels to calculate TOA fluxes for temperature, water vapour, and surface albedo responses (Soden et al., 2008; Block and Mauritsen, 2013; Shell et al., 2008; Pendergrass et al., 2018; Huang et al., 2017; Smith, 2018). Then, each flux is linearly regressed against global, annual mean surface temperature response for 150 years, where the slope of the regression line is considered the feedback value (in Wm$^{-2}$K$^{-1}$). We use the clear sky linearity test to validate the accuracy of each radiative kernel (Shell et al., 2008), where the

sum of all clear sky feedbacks is compared against the net clear sky climate feedback parameter as estimated using the Gregory regression technique with clear sky TOA flux. Three radiative kernels passed the clear sky linearity test (relative errors of less than 10%) (Figure A1), which are used to calculate an ensemble kernel mean for all feedbacks. The three sets of kernels that passed the clear sky linearity test are derived from: the Geophysical Fluid Dynamics Laboratory (GFDL) ESM (Soden et al., 2008), the Hadley Centre Global Environment Model (HadGEM2) (Smith, 2018), and from a combination of the ERA-interim

reanalysis data set and the Rapid Radiative Transfer Model (RRTM) (Huang et al., 2017).

## 2.3  Cloud Feedbacks

Cloud feedbacks cannot be calculated via the standard radiative kernel method due to nonlinearities associated with cloud vertical overlap (Soden et al., 2008). We estimate cloud feedbacks using two methods- the adjusted Cloud Radiative Effect (CRE) and cloud radiative kernel method. The CRE response is defined as the difference between clear and total sky radiative

fluxes. We adjust the CRE for the effects of environmental masking from other feedbacks using clear sky radiative kernels (Soden et al., 2004). The CRE 'adjustment' using clear sky radiative kernels takes into account differences in temperature and water vapour between a clear and cloudy atmosphere to isolate the radiative perturbation from clouds. We also account for the masking effect of $CO_2$ forcing by using a globally uniform proportionality constant of 1.16 between clear and total sky $CO_2$ forcing (Soden et al., 2008; Chung and Soden, 2015). After adjusting the CRE, the cloud flux response is regressed similarly

to noncloud feedbacks, where the slope of the regression line equals the cloud feedback. This method is performed twice to yield a value for both the shortwave (SW) and longwave (LW) cloud feedbacks.

We use cloud radiative kernels and cloud area fraction output from the International Satellite Cloud Climatology Project (ISCCP), produced from the CFMIP Observation Simulator Package (COSP) (Bodas-Salcedo et al., 2011) in CanESM2 and CanESM5, to diagnose cloud feedbacks for different cloud top pressures and optical depths (Zelinka et al., 2012a). Specifically,

we calculate a cloud area fraction response, relative to a pre-industrial control climatology, for every year, grid point, optical depth, and cloud top pressure bin for each year in the abrupt $4xCO_2$ simulation. Then, cloud radiative kernels are applied to the cloud area fraction response to derive TOA flux perturbations. Similar to noncloud feedbacks, each point is then linearly regressed against global, annual mean surface temperature response over 150 years, where the slope of the regression line is equal to the feedback value.

In this study, we consider low clouds as having their tops $\geq$ 680 hPa and non-low clouds with tops $\leq$ 680 hPa. A key limitation of COSP output is the potential obscuring of low clouds via shift in the distribution of high cloud fraction (Zelinka et al., 2018). We account for the obscuring of low clouds via normalizing low cloud fraction by upper level clear-sky fraction as in Scott et al. (2020). Using this method, non-obscured low cloud responses are weighted by the area fraction not covered by high clouds.

To further separate cloud feedbacks into contributions from cloud altitude, optical depth, and amount components, we utilize the refined decomposition technique as in Zelinka et al. (2016). Using this method, cloud kernels are decomposed into individual components for cloud amount, optical depth, altitude, and residuals, while cloud area fraction anomalies are resolved into contributions from altitude/optical depth shifts and total amount separately. For a full mathematical breakdown of this decomposition, see Appendix B in Zelinka et al. (2013) and the supplemental information document from Zelinka et al.

145 (2016).

For CanESM2, only 40 years of cloud area fraction data (years 1-20 & 120-140) were available in the abrupt-$4xCO_2$ simulation. To test the impact of sample size on our results, we subsample the output from the CanESM5 abrupt-$4xCO_2$ simulation for the same time periods as are available from CanESM2 (years 1-20 & 120-140) and find highly similar results to those obtained from the full 150 years (Figure A2). Furthermore, we find very similar results for LW and SW cloud feedback components

from CanESM2 and CanESM5 computed using the radiative kernel method and the adjusted-CRE method (Figure A3). This provides confidence that both methods are accurately capturing the pattern and magnitude of cloud feedbacks in these models.

## 3  Results

### 3.1  Effective Climate Sensitivity & Radiative Forcing

We begin by quantifying net feedback, forcing, and ECS for CanESM2 and CanESM5 (Figure 1a). Relative to CanESM2, CanESM5 has a weaker net feedback parameter (-0.64 Wm$^{-2}$K$^{-1}$) and higher ECS (5.65 K), meaning that ECS has increased by 54% between CanESM version 2 and 5. For comparison, we also show the model range of ECS for both CMIP5 and 6 using horizontal lines below the x-axis in Figure 1a, illustrating the high ECS in CanESM5 relative to all other CMIP6 models (Flynn and Mauritsen, 2020). Both versions of CanESM exhibit a strong linear relationship between surface temperature and net TOA flux (correlation coefficients are 0.92 and 0.95 for CanESM2 and CanESM5, respectively). For some ESMs, the influence of a time-varying climate feedback parameter, which could be roughly separated into a "fast response" period in the first few decades and a weaker (less negative) feedback over the latter century, had a significant influence on model's ECS values calculated via the Gregory technique (Andrews et al., 2015). Here, the strong linearity for both versions of CanESM suggests any lack of robustness in the Gregory technique is not a primary cause of the ECS increase in CanESM5.

We now turn to the different components of the forcing-feedback framework to elucidate any changes in either forcing, or feedback, and their influence on ECS. We compare the ERF for CanESM2 and CanESM5 via two methods. The ERF$_g$ is determined by the y-intercept of the Gregory regression plots (filled in circles on the y-axis in Figure 1a). The ERF$_g$ is 7.21 and 7.54 Wm$^{-2}$ for CanESM2 and 5, respectively. For comparison, we also show the ERF$_h$ as estimated using fixed-SST simulations submitted to RFMIP (open squares). Both methods produce very similar estimates of ERF— within $\pm$ 5%— which strongly suggests that the change in ECS between CanESM2 and 5 is not explained by a change in radiative forcing. We next decompose the net feedback parameter for both models to elucidate the any potential differences in the strength of radiative feedbacks.

### 3.2  Radiative Feedbacks

Planck, lapse rate, water vapour, surface albedo, and cloud TOA feedbacks are shown in Figure 1b. Planck and lapse rate plus water vapour feedbacks are roughly equal between CanESM2 and CanESM5. The surface albedo feedback is more positive in CanESM5, showing an increase of 0.05 Wm$^{-2}$K$^{-1}$ over CanESM2, which is primarily due sea ice loss over the Arctic (Swart et al., 2019), as well as a consistently more positive snow albedo feedback over polar land surfaces (not shown).

Lastly, cloud feedbacks increase in CanESM5— primarily in the SW; the result is a more positive net cloud feedback (+0.34 Wm$^{-2}$K$^{-1}$ relative to CanESM2). The net feedback (sum of all individual feedbacks) is also shown in Figure 1b to demonstrate the strong agreement the sum of kernel derived net feedback parameter (filled circles) and the net feedback parameter obtained from the Gregory regression technique (filled triangles). Strong agreement between both methods indicate that kernel ensemble mean is accurately capturing the extent of net TOA flux perturbation as outputted directly from the models.

We find our results in line with literature assessing causes behind similar increases in ECS observed by many modelling centres participating in CMIP5 and CMIP6 (Flynn and Mauritsen, 2020; Zelinka et al., 2020; Gettelman et al., 2019; Andrews et al., 2019; Golaz et al., 2019). The LW cloud feedback is positive for both versions of CanESM— increasing by 0.06 Wm$^{-2}$K$^{-1}$

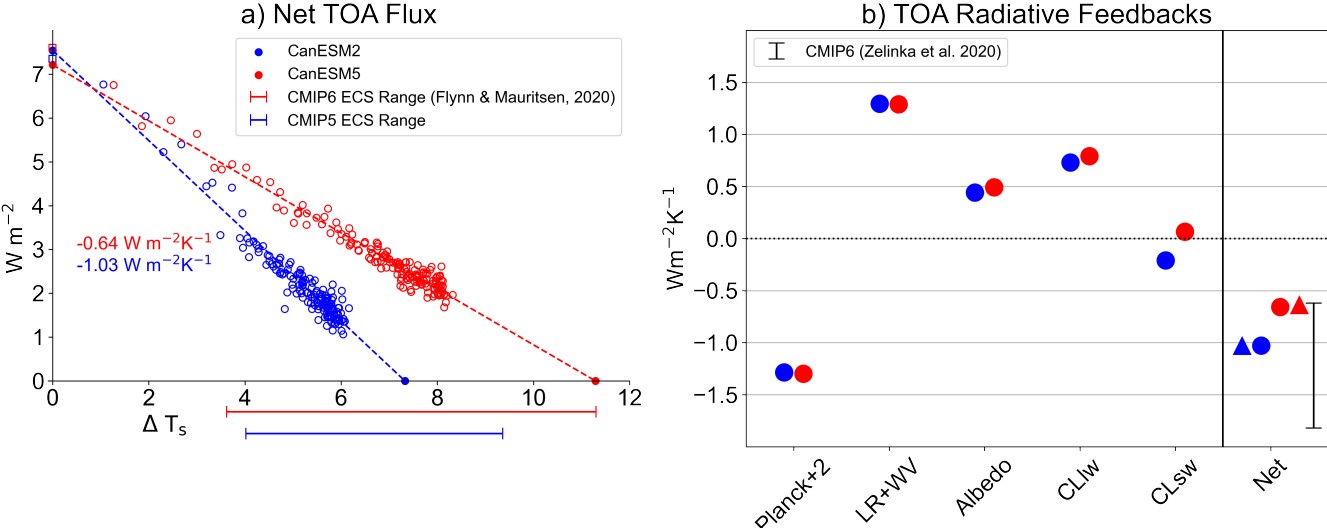

**Figure 1.** a) Net top-of-atmosphere (TOA) radiation plotted against global, annual mean surface air temperature change in abrupt-4xCO2 simulations for CanESM2 (blue) and CanESM5 (red). Standard 150 year Gregory regressions using net top-of-atmosphere radiative flux (adjusted by a preindustrial 150 year annual mean control climate) are conducted, where the x-axis intercept of the regression line divided by two is defined as the ECS, and the y-axis intercept is defined as the ERF. For comparison, the ERF, as calculated using fixed SST AMIP style runs, is shown for both versions of CanESM via the open squares along the y-axis. Bars below the x-axis denote the model range for ECS for both CMIP5 & CMIP6 (Flynn and Mauritsen, 2020). b) Global, annual mean top of atmosphere radiative feedbacks calculated using radiative kernels (in Wm$^{-2}$K$^{-1}$). From left to right, feedbacks are listed as Planck+2 (a value of 2 was added for display purposes to better illustrate differences in the other feedbacks), lapse rate plus water vapour, surface albedo, cloud, and net feedback. For comparison, we also show the net climate feedback value obtained using the standard Gregory regression approach (filled triangles), as well as the CMIP6 model range (Zelinka et al., 2020).

from CanESM2 to 5 (Figure 1b). CanESM2 exhibits a negative SW cloud feedback (-0.21 Wm$^{-2}$K$^{-1}$), while in CanESM5 the SW cloud feedback has become weakly positive (0.06 Wm$^{-2}$K$^{-1}$), indicating an absolute difference of 0.27 Wm$^{-2}$K$^{-1}$ (Figure 1b).

We quantify each individual feedback and forcing change (in terms of ECS increase) in Table 1. While the SW cloud feedback exhibits the largest difference between CanESM2 and CanESM5, both the surface albedo feedback and the LW cloud feedback offer non-negligible contributions to CanESM5's high ECS. The SW cloud feedback is the cause of at least half of the ECS increase from CanESM2 to CanESM5 (1.08 K), followed by the LW cloud and surface albedo feedbacks, respectively. Notably, There is a 3% error between the kernel ensemble-derived and model ECS values (Figure A1). As a result, we do not consider the small contributions from the Planck, and lapse rate + water vapour changes as they are close to these error bounds (Table 1). Furthermore, despite the strong linear relationship between net TOA radiation and global, annual mean surface

temperature response for both version of CanESM (Figure 1a), the regression derived ERF model difference is opposite sign of the ERF difference calculated from fixed-SST experiments (Figure 1a).

Given the importance of the cloud feedback in explaining the change in ECS from CanESM2 to 5, we devote the rest of this article to investigating the causes of this change by further decomposing both the SW and LW cloud feedbacks into their altitude, optical depth, and amount components. Although the change in net LW cloud feedback is small, we will demonstrate in the following section that this is due to compensating differences in individual components.

**Table 1.** Contributions of each component in the forcing-feedback framework to CanESM5's increased ECS (in kelvin). Individual contributions from feedbacks were calculated by substituting in feedback values from CanESM5 into CanESM2 and recalculating ECS, then taking the difference between CanESM5's ECS and the recalculated ECS. Relative contributions in parenthesis are defined as the percentage of each value of the difference between CanESM5 and CanESM2s ECS (1.98 K). This process was repeated for all individual feedbacks, as well as the ERF.

| ERF | Planck | LR + WV | Surface Albedo | LW Cloud | SW Cloud |
|---|---|---|---|---|---|
| -0.08 (-4.59 %) | 0.04 (1.96 %) | 0.08 (3.85 %) | 0.34 (17.30 %) | 0.39 (19.8 %) | 1.08 (54.38 %) |

### 3.2.1 Decomposition

The SW cloud feedback arises due to changes in cloud amount and/or optical properties. Cloud optical thickness is dependent on water path and cloud droplet size distribution (Slingo, 1989). The phase composition of clouds (liquid, ice, or mixed) is linked strongly to their optical thickness due to liquid droplets and ice crystals having different characteristic size distributions, where clouds composed of predominantly smaller liquid droplets tend to be more reflective (Pruppacher and Klett, 1980). As a result, regions where cloud composition consists entirely of liquid droplets, or are mixed phase, tend to exhibit higher albedo. In terms of feedbacks, cloud phase and amount changes have been identified as an explanatory factor to ECS spread in ESMs (Tan et al., 2016; Zelinka et al., 2016). Thus, we now decompose the cloud feedbacks using cloud radiative kernels and ISCCP simulator output for each version of CanESM following the methods described in Zelinka et al. (2012a, b) to investigate individual cloud feedback processes. We apply the decomposition separately to low and non-low clouds.

In Figure 2, we decompose LW and SW cloud feedbacks into contributions from changes in cloud optical depth, cloud altitude, and cloud amount. The LW total cloud feedback is dominated by contributions from non-low clouds (Figure 2a), with small negative contributions from low clouds (Figure 2b). Changes in cloud optical depth or amount have little radiative influence for low clouds given low cloud top temperatures are close to that of the surface, resulting in similar outgoing longwave radiation (i.e. little greenhouse effect). Furthermore, particularly for CanESM2, the strong non-low LW feedback is largely offset by a negative SW feedback of comparable magnitude. For CanESM2, contributions to the LW feedback are comparable for optical depth and amount feedbacks— with largest contribution coming from altitude. The altitude feedback arises from cloud temperature dependent emission properties, and therefore operates predominantly in the longwave for non-low clouds. Tropical free troposphere clouds rise and maintain cooler cloud top temperatures relative to the surface, thereby becoming

more efficient at trapping outgoing longwave radiation (Zelinka and Hartmann, 2010; Gettelman and Sherwood, 2016). For CanESM5, the net non-low LW feedback is approximately equal to CanESM2, albeit with a different decomposition makeup. Specifically, CanESM5 has a more positive LW altitude and optical depth feedback. However, these increases are offset by a weaker cloud amount feedback. For low clouds, the LW feedbacks are all small in magnitude. The residuals (yellow) are similarly small in both models (Figure 2), indicating that nonlinear processes are less important for understanding the changes

between models.

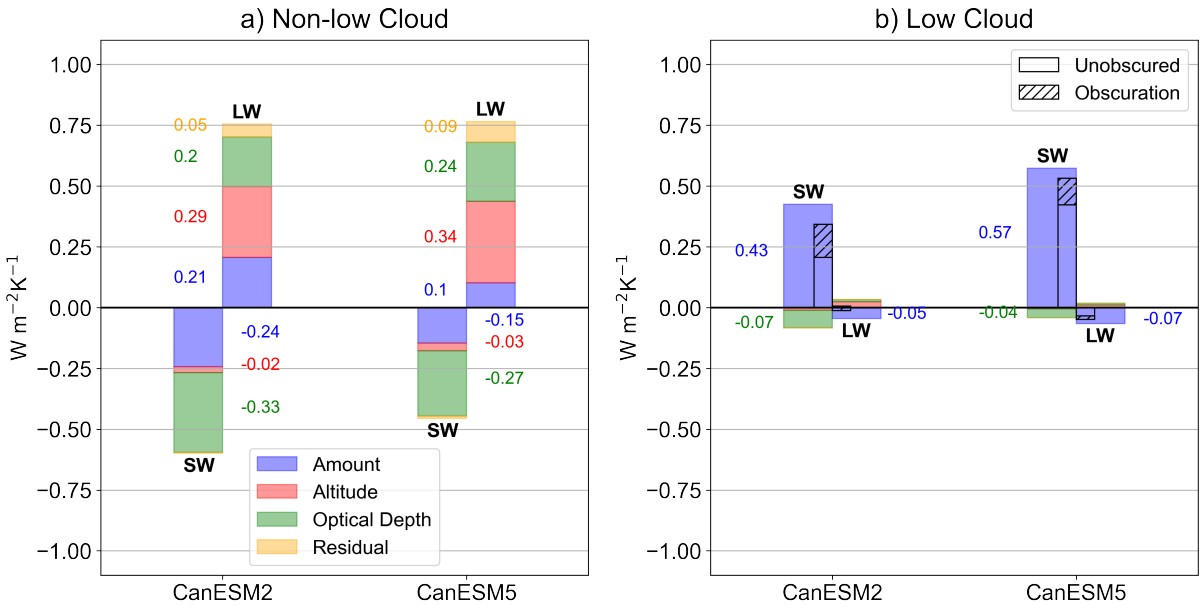

**Figure 2.** Global, annual mean decomposed cloud feedbacks for CanESM2 & CanESM5. Feedbacks are partitioned into both LW and SW contributions from cloud amount (blue), cloud altitude (red), optical depth (green), and residual (orange) terms, for non-low (panel a) and low clouds (panel b). Low cloud feedbacks are also separated via non-obscured and obscuration terms in black bars in panel b. For contributions smaller than 0.01 Wm$^{-2}$K$^{-1}$, text values were omitted for the sake of clarity.

In the SW, both models exhibit strong negative feedbacks for non-low clouds and strong positive feedbacks for low clouds (Figure 2). The negative non-low cloud shortwave feedback is driven by an increase in cloud amount and optical depth with warming. This result is consistent with both modelling and theoretical understanding of non-low cloud responses to warming. Specifically, an increasing number of liquid droplets relative to ice crystals gives rise to more mixed phase clouds, as well as

increases the proportion of liquid to ice in existing mixed phase clouds, resulting in an higher optical depth (Senior and Ingram, 1989), and a positive relationship between mid latitude cloud liquid water content and the slope of the moist adiabat as the troposphere warms (i.e. a function of temperature) (Betts, 1987). For low clouds, the SW is dominated by a strong positive amount feedback (decrease in cloudiness), with a small negative contribution from optical depth feedback (increase in cloud optical depth) (Figure 2b). For CanESM5, SW feedbacks differ from CanESM2 considerably for both non-low and low clouds.

For non-low clouds, CanESM5's optical depth feedback is weaker (less negative) than CanESM2 (+0.06 Wm$^{-2}$K$^{-1}$) (Figure 2a). While the SW non-low cloud amount feedback is also weaker in CanESM5, the difference is offset in the LW. For low clouds, the SW amount feedback exhibits the largest difference between the two model versions (+0.14 Wm$^{-2}$K$^{-1}$). The change in SW cloud feedback strength and sign between CanESM2 and CanESM5 is related to multiple feedback mechanisms operating at different cloud heights. The largest contributor is the SW low cloud amount feedback, which is more positive in CanESM5.

Changes in optical depth feedbacks are mainly important for non-low clouds, and are less negative in CanESM5.

The large difference in SW low cloud feedback strength between CanESM2 and CanESM5 raises the possibility that a portion of what the COSP interprets as a low cloud response is actually the result of changes in the spatial distribution of non-low cloud fraction under climate change; a phenomenon known as 'obscuration' (Zelinka et al., 2018). Separating for obscuration does not also separate out the amount and optical depth feedbacks. Therefore, we consider the SW unobscured low

cloud feedback as combination of changes in proportional cloud amount and optical depth. The black bars in Figure 2b indicate the both CanESM2 and CanESM5 have a non-negligible obscuration term. However, given that the extent of obscuration is similar between both versions of CanESM (see hatched bars in Figure 2b), it does not appear to be a major contributor to increased feedback strength in CanESM5 relative to CanESM2.

### 3.2.2  Spatial Distribution

Low cloud amount feedbacks are considered a robust positive feedback mechanism diagnosed from both observational and modelling studies, albeit with substantial inter-model spread in terms of strength (Eitzen et al., 2011; Clement et al., 2009; Zelinka et al., 2016). The low cloud amount feedback is closely tied to the distribution of marine stratiform cloud regimes persisting in the sub/tropics over ocean eastern boundary current regions (Klein et al., 2017). The non-low cloud optical depth feedback has been shown to have rich- and sometimes offsetting- spatial structure, owing to mostly negative feedback

mid/high latitudes and a rich zonal structure at low latitudes (Zelinka et al., 2012b, 2016). As such, we now examine the spatial distribution of both the SW low cloud amount and non-low cloud optical depth feedbacks (Figure 3).

Figure 3 shows annual mean SW non-low cloud optical depth and low cloud amount feedbacks for CanESM2 and CanESM5. The SW non-low cloud optical depth feedback is negative in CanESM2 & 5 (Figure 3a and c), with minima over the western tropical Pacific Ocean. For the SW low cloud amount feedback, both models are strongest over subtropical/tropical Eastern

Ocean basins and across the equatorial Pacific (Figure 3b and d). Notably, these regions have persistent low, stratiform cloud regimes, which are closely tied to strong temperature inversions that cap the PBL (Klein and Hartmann, 1993).

For CanESM5, increases in SW non-low cloud optical depth feedback are exhibited throughout the subtropical Pacific and tropical Eastern Pacific Ocean (Figure 3e). While there is strong positive (negative) feedback over the Eastern Indian (Western Pacific) Ocean, it is offset by a similar strength and opposing sign in the LW (not shown), and so it does not exert a major

influence on the global mean net cloud feedback (LW+SW). For SW low cloud amount feedback, CanESM5 exhibits an increase over CanESM2 in every region of persistent low cloud cover regimes, as well as across the eastern equatorial Pacific and the western ocean basin off the Brazilian coast, relating to the simulation of substantially reduced LCC under a warming climate (Figure 3f).

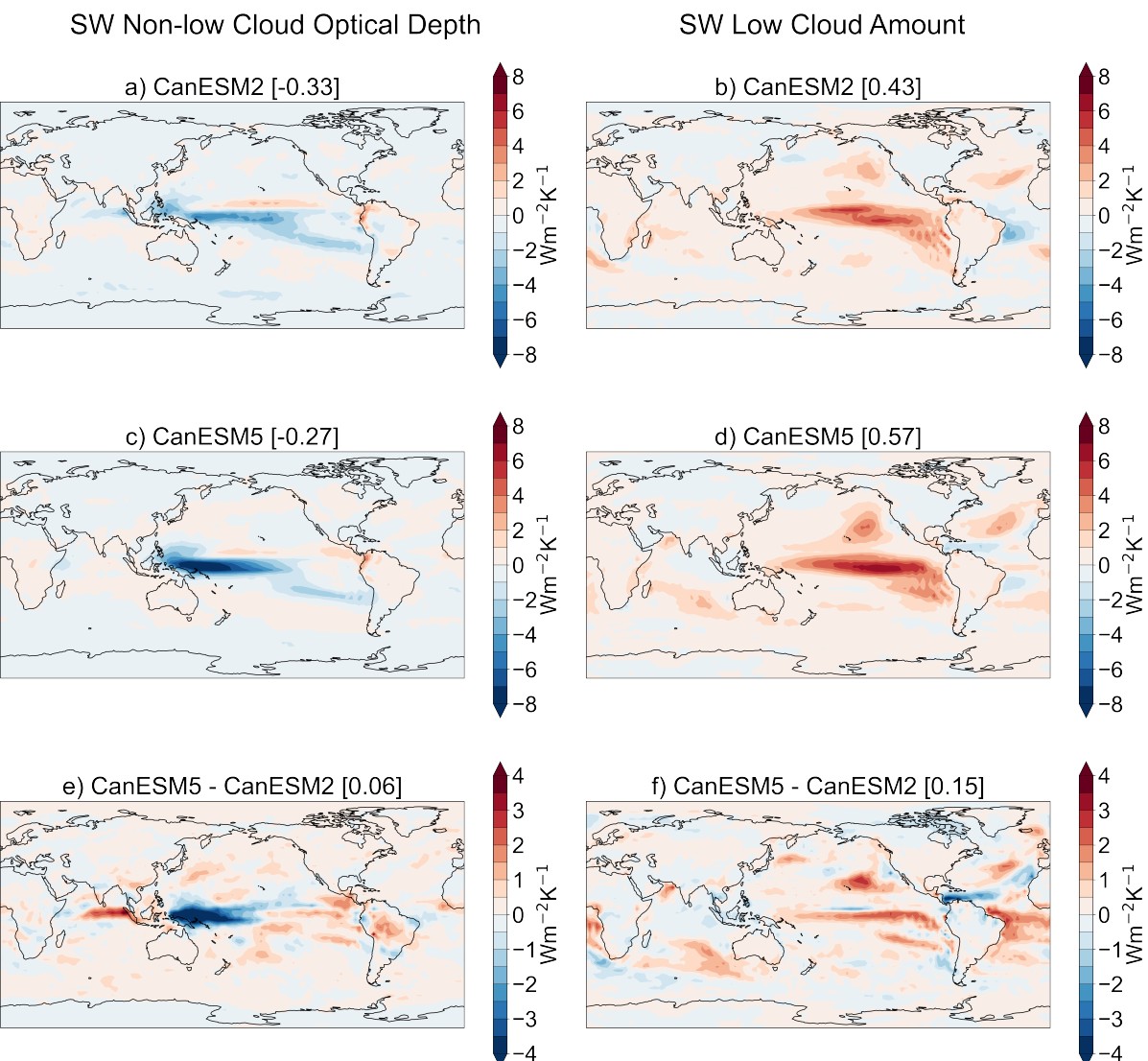

**Figure 3.** Annual mean SW non-low cloud optical depth (left) and SW low cloud amount (right) feedbacks for CanESM2 (panels a & b) and CanESM5 (panels c & d). Panels e & f show the difference between CanESM5 & CanESM2 for each respective feedback. Values in square brackets next to each subplot title denote the global mean value for each respective map. Note the difference in colour bar scales for top and bottom/middle panels.

Multiple lines of evidence from modelling studies have linked the sensitivity of LCC over the oceans to local changes in SST (Zhou et al., 2017; Andrews and Webb, 2018). Furthermore, the evolving spatial pattern of surface warming from interannual to centennial time scales is associated with differences between interannual and long term cloud feedback strength (Zhou et al., 2015). The underlying physical mechanisms linking local sea surface warming to reduced LCC are: 1) increased surface latent

heat flux dries and deepens the boundary layer via increased buoyancy-driven turbulence and resultant downward mixing from free troposphere air (Qu et al., 2015; Rieck et al., 2012), and 2) increased surface specific humidity promotes further moisture

contrast between the boundary layer and the free troposphere such that when air is mixed downward it is relatively drier (Van der Dussen et al., 2015; Qu et al., 2015). We use a proxy term, $SST^{\#}$, as a measure of distribution of surface warming in the tropics (Figure 4) (Fueglistaler, 2019). Specifically, $SST^{\#}$ is calculated as difference between the warmest 30% of tropical SSTs (i.e. the Indo-Pacific) and the tropical average, and therefore provides quantitative information about zonal asymmetries in the tropical SST pattern and how they evolve over time.

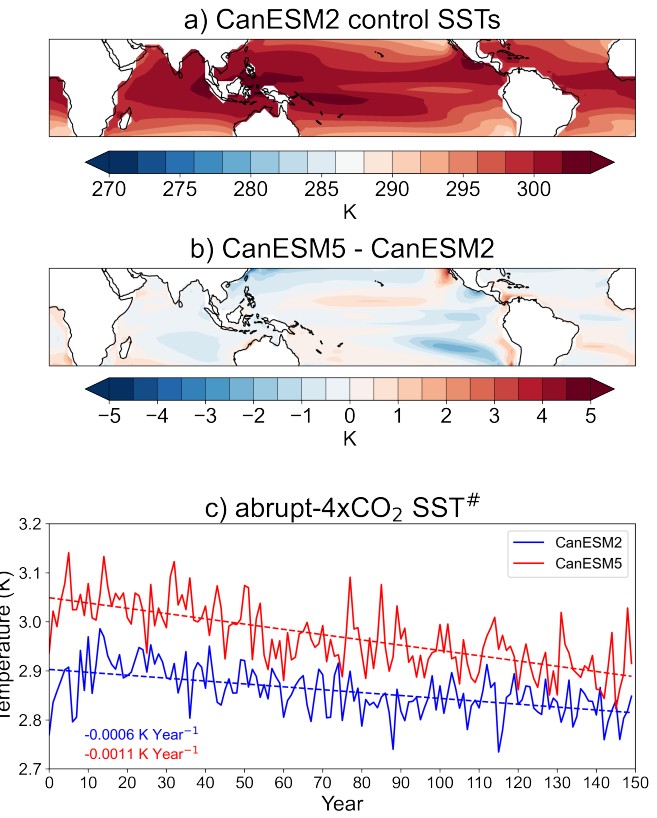

**Figure 4.** a) Pre-industrial Control mean (150 years) tropical SSTs for CanESM2. b) Pre-industrial Control mean (150 years) tropical SSTs for CanESM5, expressed as a difference relative to CanESM2. c) Annual mean time series of $SST^{\#}$ for both CanESM2 and CanESM5 for abrupt-4xCO$_2$ experiments.

The Pre-industrial control tropical SST pattern for CanESM2 is shown in Figure 4a, and exhibits a familiar zonally-asymmetric pattern with the warmest SSTs in the Indo-Pacific, and cooler waters in the east Pacific and southeastern regions of each ocean basin. Relative to CanESM2, CanESM5's control SSTs are substantially cooler— up to 3 K— over the eastern tropical Pacific (Figure 4b). Colder SSTs are also prevalent in the northern Pacific, off of the western Australian coast, and the northern tropical Atlantic ocean (Figure 4b). There is little difference between the SSTs in the Indo-Pacific, which has

important implications for SST[#]. Figure 4c shows a time series of SST[#] for abrupt-4xCO2 experiments. Given SST[#] repre-
sents the difference between the warmest waters in tropics relative to the average, a higher value indicates greater asymmetry
between warmer and cooler regions of tropical SST. The relatively colder waters in CanESM5's control climatology results
in a larger $SST_{\#}$ term, which is also illustrated at the beginning of the abrupt-4xCO2 timeseries (Figure 4c). The difference
between the models decreases as the climate warms in response to $CO_2$. The higher SST[#] term at the beginning of the sim-
ulation in CanESM5 suggests a stronger SW CRE in the pre-industrial control (Fueglistaler, 2019). The strong positive SW
cloud feedback in CanESM5, particularly from low clouds, warms the eastern Pacific (and other cooler areas) at a faster rate
than CanESM2 (Figure 3f), which gradually decreases the difference in SST[#] over the course of the abrupt-4xCO2 simulation
(Figure 4c). While the distribution of surface warming in the tropics is known to be important for low clouds (Andrews and
Webb, 2018), we also briefly analyze another important controlling factor: Estimated Inversion Strength (EIS) (Figure 5).

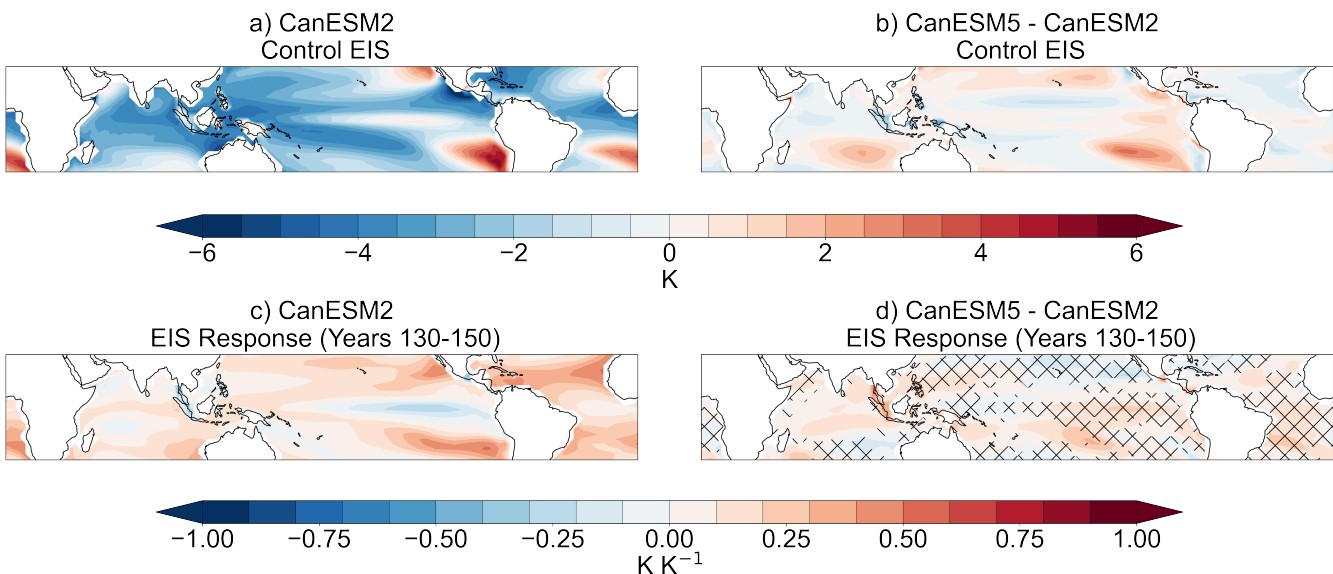

**Figure 5.** a) Annual mean control climatology (150 year mean) tropical EIS for CanESM2. b) Annual mean control climatology (150
year mean) tropical EIS for CanESM5, expressed as a difference relative to CanESM2. c) Tropical EIS response (years 130-150 mean) for
CanESM2. d) Tropical EIS response (years 130-150 mean) for CanESM5, expressed as a difference relative to CanESM2's response. For
EIS responses, each grid box value is normalized by the global mean surface temperature response (also years 130-150 mean relative to the
control period). for Hatching represents areas where the SW low cloud feedback is more positive in CanESM5.

Figure 5 shows EIS for both the control climatology and response abrupt-4xCO2 simulations, and is calculated using air
temperatures at the surface and 700hPa as in Wood and Bretherton (2006). Regions with a strong positive EIS are indicative of
a boundary layer that is decoupled from the free troposphere, with strong subsidence and cooler SSTs, and therefore more low
cloud fraction. Conversely, a strong negative EIS is indicative of a weak inversion and a surface that is coupled tightly to the
free troposhere (i.e. unstable vertical temperature profile), and is therefore inefficient at trapping moisture within the boundary

layer. In the CanESM2 control period, EIS is strongly positive across all eastern ocean basins and the eastern equatorial Pacific (Figure 5a). In CanESM5, control EIS mirrors the corresponding SST pattern (Figure 5b, 4b). In the response period, EIS increases throughout most of the tropics for both models (Figure 5c & d). CanESM5 exhibits a stronger increase in EIS throughout most of the tropics relative to CanESM2. Given that EIS and low cloud fraction are strongly positively correlated (Wood and Bretherton, 2006), a stronger response (as seen in CanESM5) would promote more cloud fraction if it were the only cloud controlling factor. However, we find no clear relationship between regions of more positive SW low cloud feedback and increased inversion strength response (Figure 5d). Inferring causal changes in cloud feedbacks between models from EIS alone is difficult given its correlation with SSTs (Scott et al., 2020) (see Figure 4b and Figure 5b). We now turn towards prescribed SST experiments from CFMIP6 to further investigate the role of SST warming distribution as a cloud controlling factor.

### 3.3 Prescribed SST Experiments

To investigate the spatial pattern of surface warming and its influence on SW cloud feedbacks, we present results from an additional experiment from the CFMIP Tier 2 experiment deck— amip-piForcing (Webb et al., 2017). The amip-piForcing experiment forces the atmosphere model with the reconstructed historical SSTs and sea ice boundary conditions from 1870 to 2014 (Hurrell et al., 2008). An important distinction from the base amip experiment is the anthropogenic and natural atmospheric forcings are held fixed to their pre-industrial control levels in amip-piForcing, which allows for a more direct comparison to abrupt-4xCO$_2$ experiments, as it removes the influence of non-SST mediated cloud responses. Thus, using amip-piForcing, we can isolate the contribution to changes in cloud feedbacks that arises due to changes in the atmosphere model (CanAM4 to CanAM5) independent of the change to the ocean model (NCAR CSM to CanNEMO). We show global mean low cloud feedbacks and SST[#] time series for both amip-piForcing and abrupt-4xCO$_2$ CanESM5 experiments in Figure 6.

For low clouds, the SW feedback is stronger (+0.21 Wm$^{-2}$K$^{-1}$) in the abrupt-4xCO$_2$ experiment than amip-piForcing— largely due to the cloud amount feedback, but with a smaller contribution from the optical depth feedback (Figure 6a). This result is consistent with theoretical understanding relating warmer local SSTs to reduced boundary layer marine cloud cover as outlined in the previous section, which is further emphasized by the geographic distribution of warming in the amip-piForcing experiment (Figure 6b). In amip-piForcing, the eastern Pacific warms only by a small amount relative to the abrupt-4xCO$_2$ experiment (Figure A4), which is exemplified by the SST[#] time series. Furthermore, the SST[#] trend is opposite in amip-piForcing, illustrating the warming in the Indo-Pacific warm pool over the historical record (Figure A4), which is not present in coupled model abrupt-4xCO$_2$ experiments (Andrews et al., 2018).

We summarize feedback results from all experiments considered in this study in Table 2. The Planck response is nearly identical for all three experiments. The combined lapse rate and water vapour feedback is similar in abrupt-4xCO$_2$ experiments for both models, but noticeably more negative in the amip-piForcing experiment. The combined result is a product of the more negative lapse rate feedback, which arises due to the relatively stronger warming in the Indo-Pacific warm pool (Andrews et al., 2018) from the reconstructed SST dataset used as forcing (Hurrell et al., 2008). Surface temperatures in the deeply convective warm pool increase Earth's emission temperature via enhanced latent heat release, which stabilizes the vertical temperature profile and warms the upper troposphere. Outgoing longwave radiation increases and produces a strong radiative cooling effect.

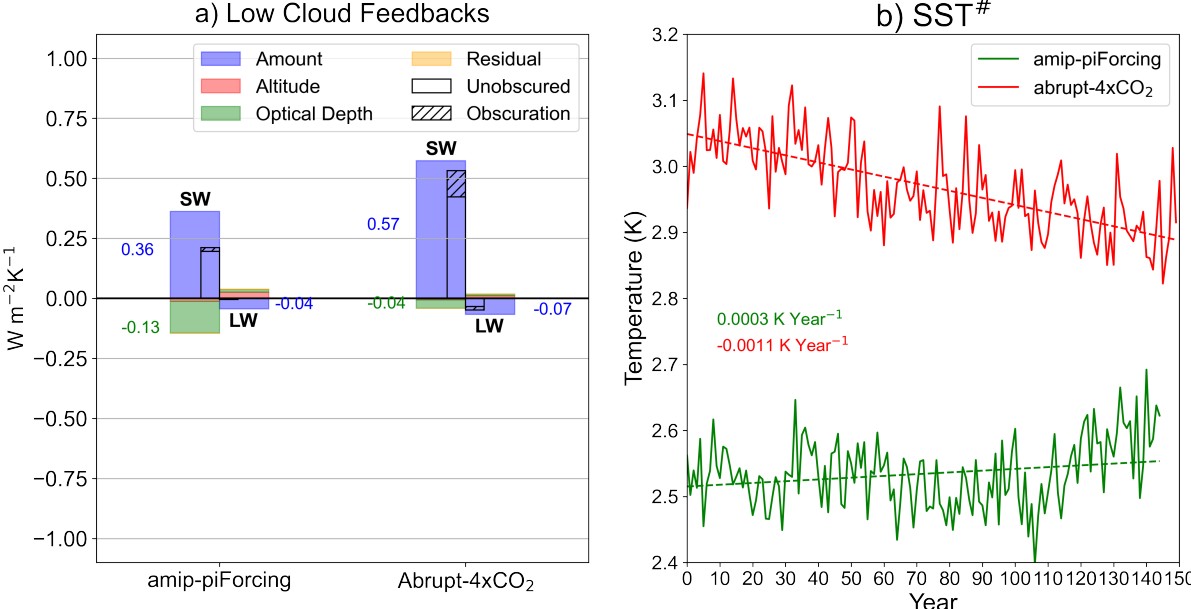

**Figure 6.** a) Global, annual mean decomposed cloud feedbacks for CanESM5's amip-piForcing and abrupt-4xCO$_2$ experiments. Feedbacks are partitioned into both LW and SW contributions from cloud amount (blue), cloud altitude (red), optical depth (green), and residual (orange) terms. Feedbacks are also separated via non-obscured and obscuration terms in black bars. For contributions smaller than 0.01 Wm$^{-2}$K$^{-1}$, text values were omitted for the sake of clarity. b) Annual mean time series of SST$^{\#}$ for both CanESM5 experiments.

The surface albedo feedback is stronger in the CanESM5 for coupled model experiments, but reduced in amip-piForcing

relative to CanESM5's abrupt-4xCO$_2$ experiment, which is likely a result of the constrained sea ice boundary conditions present. The longwave cloud feedback is similar across all three experiments/models, and the SW cloud feedback has strongly increased in CanESM5 for coupled experiments, but is reduced to a similar magnitude to CanESM2 abrupt-4xCO$_2$ in the amip-piForcing experiment. The net feedback result for the amip-piForcing experiment is strongly negative in CanESM5 relative to abrupt-4xCO$_2$, which is due to the combined effect of a reduced surface albedo, SW cloud, and lapse rate feedbacks.

## 4   Discussion & Conclusions

In this study, we have analyzed both forcing and feedback in idealized experiments with instantaneous quadrupling of atmospheric CO$_2$ (abrupt-4xCO$_2$) using two versions of CanESM to elucidate the underlying cause behind CanESM5's increased ECS (5.65 K). Using radiative kernels and output from RFMIP, we find only modest differences in both forcing and non-cloud feedbacks, with a small contribution from a slight increase in the surface albedo feedback. The largest difference in feedback

strength between CanESM2 to CanESM5 is from the cloud feedback, particularly in the SW. Further breakdown of the cloud feedback into its individual components (optical depth, altitude, and amount) at distinct cloud top heights (<680 hPa for non-

**Table 2.** Summary of radiative feedbacks, calculated using a combined radiative kernel/regression method (adjusted-CRE in the case of the cloud feedbacks listed here) for both model version abrupt $4xCO_2$ experiments, as well as CanESM5's amip-piForcing experiment. All feedbacks are in units of $Wm^{-2}K^{-1}$. Feedbacks from the amip-piForcing run were calculated using the 1980-2010 period as a baseline

| Model | Planck | LR + WV | Surface Albedo | LW Cloud | SW Cloud | Net |
|---|---|---|---|---|---|---|
| CanESM2 ($4xCO_2$) | -3.29 | 1.26 | 0.44 | 0.73 | -0.21 | -1.03 |
| CanESM5 ($4xCO_2$) | -3.30 | 1.29 | 0.49 | 0.79 | 0.06 | -0.66 |
| CanESM5 (pi-Forcing) | -3.31 | 0.96 | 0.45 | 0.76 | -0.21 | -1.35 |

low, $\geq 680$ hPa for low) revealed that the SW low cloud amount and non-low cloud optical depth feedbacks are the dominant contributor to CanESM5's increased ECS (+0.14 and 0.06 $Wm^{-2}K^{-1}$, respectively) in abrupt 4x-$CO_2$ simulations. Analysis of the spatial pattern for each feedback showed the largest model differences in SW low cloud amount feedback over subtropical eastern ocean basins and across the equatorial Pacific ocean, and in SW non-low optical depth feedback over the subtropical and extratropical Pacific ocean.

We analyzed the spatial pattern of surface warming and its influence on the SW low cloud feedback using the CFMIP tier 2 amip-piForcing experiment in CanESM5, which exhibited significantly reduced SW cloud feedback due to the lack of local warming in regions with persistent low cloud cover— in agreement with studies linking warmer (colder) SSTs to decreased (increased) LCC (Qu et al., 2014; Bretherton and Blossey, 2014; Brient and Schneider, 2016). We found a similar strength in LW cloud feedback from both abrupt-4x$CO_2$ and amip-piForcing experiments, and a reduced lapse rate feedback in the amip-piForcing experiment due to the relatively stronger surface warming in the Indo-Pacific warm pool. While lacking an analogous amip-piForcing experiment for CanESM2, the results presented here agree with similar experiments conducted using other ESMs (i.e. more negative lapse rate and SW cloud feedback) that have been studied with respect to pattern effects of warming (Andrews et al., 2018). Furthermore, the SW cloud feedback strength in the CanESM2-abrupt-4x$CO_2$ is equal to that of the CanESM5 amip-piForcing experiment despite a different pattern effect of warming (Figure 4a & Figure 6b). The amip-piForcing results presented here confirm the well documented relationship between local SSTs as a controlling factor for low clouds. This result suggests that both the pattern of warming itself, and the sensitivity of low cloud fraction to this pattern plays a key role in CanESM5's higher ECS. Disentangling the role of the ocean model replacement (The National Centre for Atmospheric Research CSM ocean model for CanESM2 (Gent et al., 1998) to CanNEMO for CanESM5 (Swart et al., 2019)) and the developmental changes to cloud microphysics in CanAM5 (e.g. the new autoconversion scheme and aerosol indirect effect) is a subject for future work.

Our results add further evidence the recent trend of several ESMs participating in CMIP6 exhibiting higher ECS than their CMIP5 counterpart— predominantly due to changes in SW cloud feedback strength and/or aerosol-cloud interactions
(Gettelman et al., 2019; Andrews et al., 2019; Bodas-Salcedo et al., 2019). However, it is worth noting that several modelling centres report increased ECS sourced from distinct developments in newer versions of their respective model (e.g. addition of a mixed-phase cloud scheme and improved aerosol-cloud interactions, as well as the higher horizontal ocean model resolution its influence on SSTs in cold upwelling regions, in HadGEM3 (Bodas-Salcedo et al., 2019; Mulcahy et al., 2018; Andrews et al., 2019).

Finally, we emphasize that the results presented in this study do not seek to comment on the plausibility of climate sensitivity from either version of CanESM. Recently, there has been an expansion of work relating constraints on climate sensitivity through the use of the satellite and paleoclimate observational records (Sherwood et al., 2020). Furthermore, there are limitations of interpreting the validity of climate sensitivity results (as calculated here) due to uncertainties associated with statistical methods (e.g. assuming a time-invariant climate sensitivity parameter via the regression approach) (Gregory et al., 2004).
However, we reiterate the scope of this study: establishing a causal link for the increased climate sensitivity from CanESM2 to CanESM5 under long term, idealized climate change.

*Code and data availability.* All model output from both versions of the Canadian Earth System Model analyzed in this study are publicly available for download via the Earth System Grid Federation (https://esgf-node.llnl.gov/projects/esgf-llnl/). Source code for the Canadian Earth System model can be found at https://gitlab.com/cccma/canesm. The particular model version (5.0.3) that contributed output to CMIP6
is available from https://doi.org/10.5281/zenodo.3251113. Code used for analysis of model output and production of figures located at https://github.com/JohnVirgin/GMD_CanESM_Clouds.

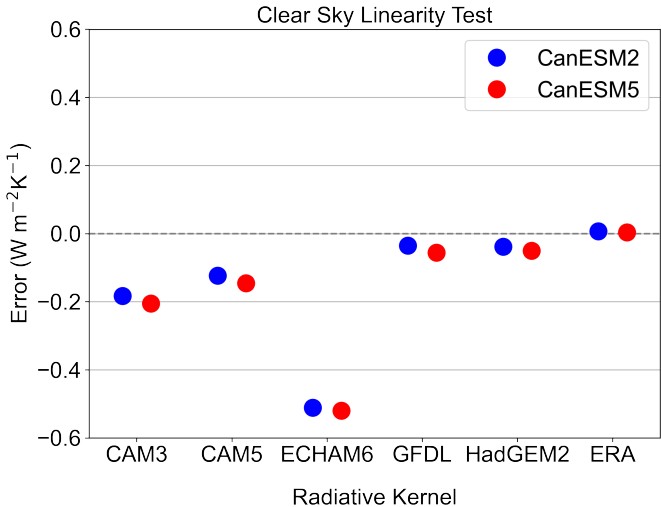

**Figure A1.** Clear sky linearity test for 6 sets of radiative kernels considered in this study (CAM3, CAM5, ECHAM6, HadGEM2, and ERA kernels) tested using each version of CanESM. Y-axis error is defined as the absolute difference between the Gregory regression derived net clear sky climate feedback parameter, and radiative kernel derived net clear sky climate feedback parameter.

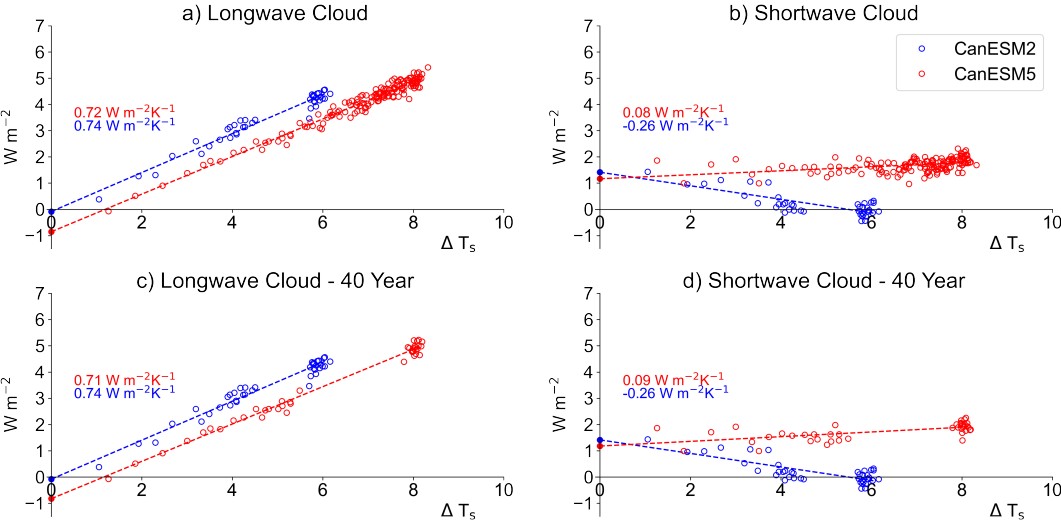

**Figure A2.** Cloud long- and shortwave flux plotted against global, annual mean surface temperature change in abrupt-4xCO2 simulations for CanESM2 (blue) and CanESM5 (red), calculated using the cloud radiative kernel method. Standard 150 year Gregory regressions are conducted, where the slope of the regression line equals the cloud feedback (in $Wm^{-2}K^{-1}$). Panels a & b show regressions using all available years of data for each model version, whereas panels c & d show subsampled data for CanESM5 (years 1-20 & 120-140).

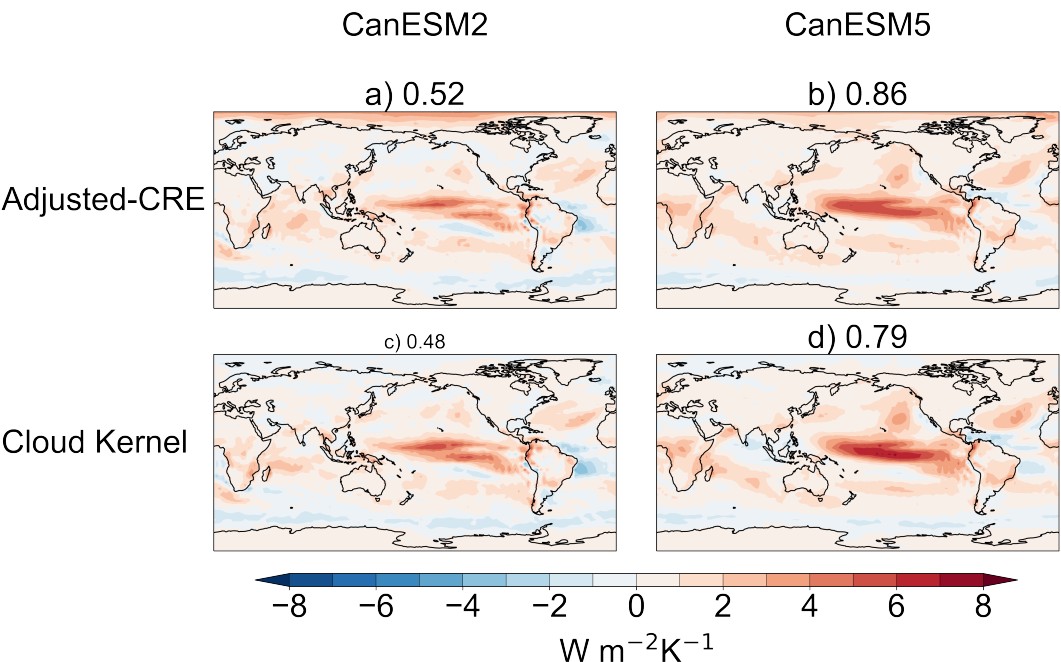

**Figure A3.** Comparison of annual mean net cloud feedbacks for CanESM2 (panels a & c) and CanESM5 (panels b &d), calculated using the adjusted-CRE method and the cloud kernel method. Global mean values are shown in square brackets next to each subplot title. CanESM2 Pearson's $r = 0.72$ ($p < 0.01$); CanESM5 Pearson's $r = 0.86$ ($p < 0.01$).

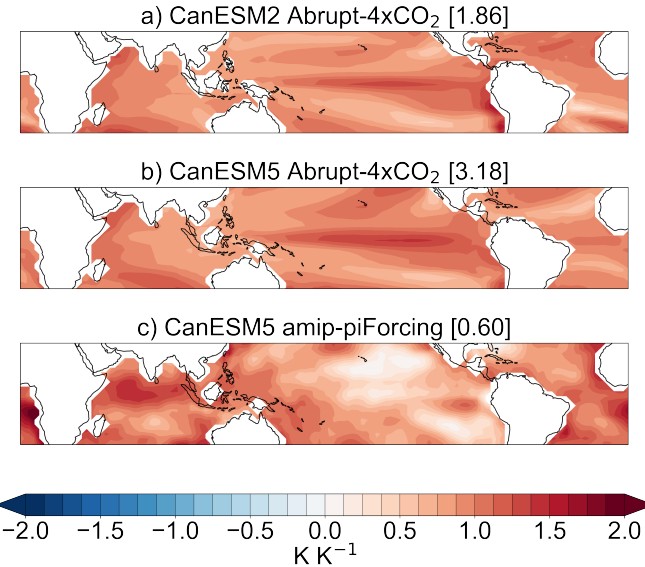

**Figure A4.** Tropical SST response for the a) CanESM2 abrupt 4xCO₂ simulation, b) CanESM5 abrupt 4xCO₂ simulation, and c) CanESM5 amip-piForcing simulation. Responses are defined as the difference between 20 year means taken from the beginning and end of each simulation. All grid box values are divided by the global mean response for each respective simulation, which is shown in square brackets in each subplot title.

*Author contributions.* JGV performed analysis on model output and created all figures used in this study. JGV, CGF, JC, KVS, and TM wrote the manuscript.

*Competing interests.* The authors declare that there are no competing interests.

*Acknowledgements.* The authors acknowledge Mark Zelinka, Neil Swart, and Nathan Gillet for providing useful insights and comments that helped improve this study. The authors also acknowledge the World Climate Research Programme's (WCRP) Working Group on Coupled Modelling (WGCM) and the Earth system Grid Federation (ESGF) for their role in providing access to the data used in this study.

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
