# Peer review of "Cloud Feedbacks from CanESM2 to CanESM5.0 and their Influence on Climate Sensitivity"

_Geoscientific Model Development, 2021_

## Referee Comment (RC2)

Comments on "Cloud Feedbacks from CanESM2 to CanESM5.0 and their
Influence on Climate Sensitivity"

This paper compared the radiative forcing and feedback between CanESM2 and CanESM5 and
tried to understand what causes the higher ECS in CanESM5. In the fully coupled experiments
(abrupt-4xCO2), they found the cloud feedback is the dominant factor, especially the shortwave
component, which is consistent with previous findings. Further decomposition of cloud feedback
using the cloud radiative kernel method showed the reduced shortwave low cloud amount and
non-low cloud optical depth are two important factors to produce stronger positive cloud
feedback. The cloud feedbacks from amip-p4K and amip-future4K were discussed to
understand the impact of spatial sea surface patterns.
The topic is quite important to help understand the high ECS from CMIP5 to CMIP6 given that
CanESM5 shows the highest ECS. The motivation and method description are clear and
easy-following, however, I think results are not appealing and deep enough to extend and
broaden the current understanding about stronger positive cloud feedback because many
changes reported in the paper are lacking solid physical explanations from their model
perspective. For example, many high ECS models show stronger positive cloud feedback. I
think the more appealing question is: why is the positive cloud feedback getting so strong from
CanESM2 to CanESM5, not only showing from those cloud feedback decompositions? So I
would recommend making substantial revisions on the analysis and explanation to the finding
before publishing the paper to GMD.

Major comments:
1.  I think it's not enough to just show the cloud feedback difference between CanESM2 and
    CanESM5. A more physical explanation is expected, instead of only presenting those
    changes and referring to previous studies. I list some questions/comments below and I
    think they need to be addressed more. I don't list all of them here and I think
    explanations to those significant signals are the key point I would like to see.
    1.  L177-178: the motivation to show LW components is because although the
        difference between CanESM2 and CanESM5 is small, there is an important
        compensation between its individual components. However, in the following
        discussion, e.g. L198-200, I don't see many discussions about these
        compensations.
    2.  L205-210: Authors refer to previous proposed mechanisms about increased
        cloud liquid water in mixed-phase clouds or increased cloud liquid water content
        and the slope of the moist adiabat, which could explain the negative non-low
        cloud optical depth feedback. Why not further examine these particular state
        variables to check whether model response is consistent with the previous
        studies? Furthermore, the negative mid-latitude cloud optical depth feedback is
        quite popular in climate models, and I think the more important question is:
        instead of the feedback sign, what controls the feedback magnitude change from
        CanESM2 to CanESM5?
    3.  L236: "For the SW low cloud amount feedback, …" I notice the strongest SW low
        cloud amount feedback is in the central to eastern tropical Pacific, rather than

those regions with highest EIS over marine low cloud regions. If you think this low cloud amount change is controlled by the EIS or other factors, it might be better to dig it out and give more physical explanations.

4. Table 1: 1) amip-future4K is closer to the abrupt-4xCO2 feedback for non-low clouds, but amip-p4K is closer to the coupled for low clouds. Why? 2) For the total cloud feedback, amip-p4K is closer to coupled for CanESM2. amip-future4K is closer to coupled for CanESM5. Why?

5. Figure 5: 1) non-low cloud: LW and SW cloud amount feedback is largely reduced in CanESM5. Also, SW cloud optical depth feedback is reduced from -0.46 to -0.35 W/m$^2$/K. SW cloud amount feedback is reduced from -0.3 to -0.09 W/m$^2$/K. This change is more significant in amip-p4K and amip-future4K than coupled (Figure 3). Why? I expect the text will show some discussions about this.

2. I would suggest revising some figures or contents to convey clearer ideas.
    1. Section 3.3: I don't think this section gives a good explanation of the impact of the spatial pattern of SST change. Except for those changed cloud feedback components, I think more discussion and analysis to understand these changes are more important. Also, comparison with abrupt-4xCO2 results would be also interesting.
    2. Figure 2: I think this figure is not more informative than just adding the global-mean SW and LW cloud feedbacks in Figure 1 as another two columns. As for the uncertainty brought by different radiative kernels, I think they are not the key point here. So, except for the uncertainty consideration, I think Figure 2 is not necessary to be only used to show the LW and SW cloud feedbacks.
    3. Figure 6: I am not able to get quite clear information from this figure if I don't read the text. First, I suggest that you should report the significance of the correlation. Second, after reading the text, I think your motivation is to show the relationship between SST pattern anomaly and feedback anomaly, but this has been widely investigated in previous studies. I am not sure whether it's necessary to have this figure here to confirm this. Additionally, I think more comparisons of cloud feedbacks from amip4K/amip-future4K and abrupt-4xCO$_2$ experiments will be better because it can reveal how largely the amip-type feedbacks could reproduce the coupled feedbacks in this model and what contributes to these differences.

Minor comments:
    1. In section 2.1 describing the model, it is better to document the major difference between these two models because you are discussing their difference.
    2. L74: what does 'in part-' mean?
    3. L80: should be 'sstClim and sstClim4xCO$_2$' in the bracket rather than 'piControl and abrupt4xCO$_2$'?
    4. L99: It might be better to list these three kernels that passed the clear-sky linearity test and the reader doesn't need to find Figure A1 if they are curious what they are.
    5. L108: "global uniform proportionality constant" -- why not showing the value here?
    6. L144: "strong linear relationship between surface temperature and net TOA flux" → why?

7. L157: suggest changing "due to increases over the Arctic from sea ice loss" to "due to the sea ice loss over the Arctic"
8. Figure 1: why not also showing LW and SW cloud feedbacks in panel (b)? That will be more direct for readers to see the relative contribution of SW and LW.
9. L173: I think it should be "Figure 2" not "Figure 3" in the bracket.
10. L209: "The situation for the non-low …" -- This sentence is not clear to me. Please consider rephrasing.
11. Table 1 caption: should be "amip-future4K" not "amip-future-4K"

---

## Author Comment (AC1)

**Author's Statement**

We thank both reviewers for their incredibly thorough and insightful comments of our initial submission. As a response, we have made a significant number of major revisions to the results and discussion to better illustrate the value of our analyses. Both reviewers expressed concerns over the way our amip results were presented (Section 3.3). Namely, the lack of clarity and cause of differences between amip versus coupled model results, as well as using them to contextualize the SW low cloud feedback from a physical science perspective. We agree that this was not clear in our first submission and have decided to overhaul the final section by using a separate amip experiment instead. We have decided to not consider comparing cloud feedbacks between the amip-p4K/future4K experiments with abrupt-4xCO$_2$ runs given the presence of time varying forcings in amip experiments, which were not present in the coupled model simulations.

We also recognize that many studies have been done assessing shifts in cloud feedbacks amongst CMIP6 models, and the appealing question of addressing *why* from a model development perspective. However, we believe that the attribution of cloud feedback differences to specific model developments to be outside the scope of this work (albeit certainly a topic for future work). We believe that there is still value in the analysis we present as it allows for a more granular, direct comparison between two model versions, whereas much of the work done on this subject (outside of modelling groups themselves) scopes their work around the CMIP6 ensemble. Therefore, we have decided to expand the first result section to better clarify the role of other feedbacks and their direct contribution to the ECS shift from CanESM2 to 5 (see Table 1), and to expand our analysis on a few cloud controlling factors and their evolution in the abrupt-4xCO$_2$ experiments for both models (see revised Figures 4 and 5).

**Reviewer 1 Comments and Responses**

**Perhaps my main criticism is the explanation of differences in SW low cloud feedback between amip-p4K and amip-Future (Figure 6, L271-280). I think the discussion speculative and not very convincing, as one cannot clearly see how the population of feedback and pattern anomalies is distributed.**

Thank you for your comment. We have made major revisions to the final section to better explain the role of local SSTs to the SW low cloud feedback. All figures pertaining to the amip-p4K/future4K results have been removed, and we have performed new analysis on CanESM5's amip-piForcing experiment from CFMIP6. See L303-343, along with the revised Figure 6 and Table 2.

**Figure A3: given that this figure is referenced before Figure 2, I'd recommend using the full figure caption, not "As in Figure 2".**

Thank you for your suggestion. Figure 2, as referenced from this comment, has been removed. Figure A3, which is now Figure A2 in the revised document, has been revised according to your suggestion.

**Figure A2: This figure is referenced after Figure A3, I suggest swapping the order. Figures: Please increase text size of axes titles and tick labels.**

Thank you for your suggestion. These figures have been swapped and figure tick/title/label sizes have been increased for all figures.

**L17: Can the Chaney models be considered ESMs?**

Good point. We've clarified the distinction between the Charney models and modern ESMs. See L17.

**L24-25: - "particularly with regards to properties such as cloud optical depth, which reflect shortwave (SW) radiation and cool the planet". This reads weird, clouds reflect radiation, cloud optical depth is a property that changes how much it is reflected.**

Thank you for pointing this out. We have revised the line to clarify this point. See L26-27.

**L43: (CanESM2). (Flynn and Mauritsen, 2020). I'd suggest re-arranging this sentence to avoid model acronym and the citation being together.**

We have revised these lines accordingly. See L46-47.

**L70-71: Strictly speaking, ECS doesn't need the definition of a CFP (time-varying or not). One can run a model to equilibrium (with a practical definition of "equilibrium") and calculate ECS without invoking a feedback parameter.**

Thank you for your comment. We have revised these lines to clarify that true Equilibrium Climate Sensitivity is not dependent on defined CFP and can be simulated using models, thus not requiring any form of statistical extrapolation. See L81-85.

**L165-167: optical thickness and emissivity are not the only properties, cloud fraction and top pressure/temperature can produce radiative feedbacks (as discussed later in the paper).**

Thank you for your comment. We have removed this line as the revised document, and the beginning of section 3.2.1 (L202) documents the various physical mechanisms behind decomposed cloud feedbacks.

**L169-174: Figure 3 -> Figure 2**

This has been corrected given Figure 2 from the initial submission has been removed and its content folded into Figure 1b.

**L193-194: Perhaps worth noting that, at the same time, the SW feedback (especially for CanESM2) shows differences of similar magnitude in the opposite direction.**

We have added a line to note the SW offset from the anomalously strong LW non-low Cloud feedback in CanESM2. See L215.

**L195-198: I think the term 'emissivity' is wrongly used here (perhaps you mean emission). As explained in the second sentence, the altitude feedback is related to the temperature difference between cloud tops and the surface. That mechanism doesn't need a change in cloud emissivity.**

Thank you for your comment. We've revised this line accordingly. See L218.

**L311: The new developments are included in HadGEM3-GC3.1.**

We have included the additional citation and point regarding HadGEM3. See L372-384.

**Reviewer 2 Comments and Responses**

**L177-178: the motivation to show LW components is because although the difference between CanESM2 and CanESM5 is small, there is an important compensation between its individual components. However, in the following discussion, e.g. L198-200, I don't see many discussions about these compensations.**

Thank you for your comment. Our goal for motivation the decomposed LW components was to clarify that this is due to compensations in CanESM5 that create an offset. We have revised and clarified this point regarding the a slightly weaker amount feedback, which offsets the small increases in non-low altitude and optical depth feedbacks in CanESM5. See L220-223.

**L205-210: Authors refer to previous proposed mechanisms about increased cloud liquid water in mixed-phase clouds or increased cloud liquid water content and the slope of the moist adiabat, which could explain the negative non-low cloud optical depth feedback. Why not further examine these particular state variables to check whether model response is consistent with the previous studies? Furthermore, the negative mid-latitude cloud optical depth feedback is quite popular in climate models, and I think the more important question is: instead of the feedback sign, what controls the feedback magnitude change from CanESM2 to CanESM5?**

Thank you for your comment. We agree that the question of what is controlling the magnitude of feedback change to be especially salient. However, we believe that exploring this question from a development perspective is be outside the scope of work presented here. Our goal was to provide a granular look at differences in feedbacks between both model versions, as well as to explore the role of a new ocean model and the potential influence of the pattern effect on CanESM5's SW cloud feedback. To this end, we have added extra analysis and a table in section 3.2 (see L188-196 and Table 1) and another figure (See Figure 4) to better illustrate the spatial evolution of SSTs in both models over the course of abrupt-4xCO$_2$ experiments.

**L236: "For the SW low cloud amount feedback, …" I notice the strongest SW low cloud amount feedback is in the central to eastern tropical Pacific, rather than those regions with highest EIS over marine low cloud regions. If you think this low cloud amount change is controlled by the EIS or other factors, it might be better to dig it out and give more physical explanations.**

Thank you for your comment. To address this, have expanded our analysis on both the two primary controlling factors for low clouds in the abrupt-4xCO$_2$ experiments— EIS and SSTs. Specifically, we analyze the evolution of SST changes through the tropics (See Figure 4) as well as the control climatology EIS and warming response (Figure 5).

**Table 1: 1) amip-future4K is closer to the abrupt-4xCO2 feedback for non-low clouds, but amip-p4K is closer to the coupled for low clouds. Why? 2) For the total cloud feedback, amip-p4K is closer to coupled for CanESM2. amip-future4K is closer to coupled for CanESM5.**

**Figure 5: 1) non-low cloud: LW and SW cloud amount feedback is largely reduced in CanESM5. Also, SW cloud optical depth feedback is reduced from -0.46 to -0.35 W/m2/K. SW cloud amount feedback is reduced from -0.3 to -0.09 W/m2/K. This change is more significant in amip-p4K and amip-future4K than coupled (Figure 3). Why? I expect the text will show some discussions about this.**

Thank you for your comments regarding these results. We agree that there are interesting differences in the decomposition between the amip and coupled model results (as you have outlined). As noted in our summary above, we have made the decision to overhaul the final section, which included our amip-p4K/future4K results to better illustrate the role of local SSTs on the SW low cloud feedback, and therefore

ECS, in CanESM5. After some consideration, we decided that the amip-p4K and future4K simulations were not appropriate for addressing our science question given they use time varying atmosphere forcings (e.g. aerosols from 1970-2014). Given the cloud response in these amip experiments could also be a response to forcing, comparing the amip results to coupled model results, which use the pre-industrial climate state (except for quadrupling $CO_2$), could obfuscate feedback-induced responses from forcing responses.

As a result, we have revised our analysis to use the amip-piForcing experiment from the CFMIP contribution to CMIP6. We believe this provides a more direct analogue for exploring the role of constrained SSTs, and therefore SW cloud feedback, in CanESM5. Section 3.3 (L310-339 and Figure 6/Table 2) contains all new results pertaining to this change.

**Section 3.3: I don't think this section gives a good explanation of the impact of the spatial pattern of SST change. Except for those changed cloud feedback components, I think more discussion and analysis to understand these changes are more important. Also, comparison with abrupt-4xCO2 results would be also interesting.**

Thank you for your comment. As mentioned above, we have completely revised this section to better address this comment. Furthermore, we have included extra analysis on the spatial pattern of SST change for the abrupt-4xCO$_2$ experiments from L276-293, as well as the addition of Figure 4.

**Figure 2: I think this figure is not more informative than just adding the global-mean SW and LW cloud feedbacks in Figure 1 as another two columns. As for the uncertainty brought by different radiative kernels, I think they are not the key point here. So, except for the uncertainty consideration, I think Figure 2 is not necessary to be only used to show the LW and SW cloud feedbacks.**

Thank you for this suggestion. We agree, and have removed Figure 2 and folded its results into Figure 1, which now shows the cloud feedback decomposed into its LW and SW components.

**Figure 6: I am not able to get quite clear information from this figure if I don't read the text. First, I suggest that you should report the significance of the correlation. Second, after reading the text, I think your motivation is to show the relationship between SST pattern anomaly and feedback anomaly, but this has been widely investigated in previous studies. I am not sure whether it's necessary to have this figure here to confirm this. Additionally, I think more comparisons of cloud feedbacks from amip4K/amip-future4K and abrupt-4xCO2 experiments will be better because it can reveal how largely the amip-type feedbacks could reproduce the coupled feedbacks in this model and what contributes to these differences.**

Thank you for these suggestions. We agree, and made the decision to remove this Figure and not try to replace it in the overhauled section 3.3, as we felt it did not aid in conveying the point we were trying to make regarding the influence of local warming on low clouds.

**In section 2.1 describing the model, it is better to document the major difference between these two models because you are discussing their difference.**

Thank you for this suggestion. We have revised section 2.1 to include a brief discussion of differences between the two model versions. See L53-68.

**L74: what does 'in part-' mean?**

Partially. We have revised this line to clarify this point. See L86-88.

**L80: should be 'sstClim and sstClim 4xCO2' in the bracket rather than 'piControl and abrupt 4xCO2'?**

Thank you for pointing this error out. We have revised this line accordingly. See L93.

**L99: It might be better to list these three kernels that passed the clear-sky linearity test and the reader doesn't need to find Figure A1 if they are curious what they are.**

We have expanded the discussion on the clear sky linearity test to include mentions and citations of the radiative kernels that passed the test. See L112-115.

**L108: "global uniform proportionality constant" – why not showing the value here?**

Thank you for pointing this out. We have added a line stating the value (1.16 between clear and total sky $CO_2$ forcing) on L123.

**L144: "strong linear relationship between surface temperature and net TOA flux" → why?**

Our rationale for pointing this result out was to note that, for many ESMs, the FNET vs surface temperature response in abrupt-4x$CO_2$ experiments exhibits distinct CFPs for the intial response (first few decades) and the long term response (last 100-120 years). Specifically, a weaker CFP in the long term response. This so called "kink" in such Gregory regression plots is absent for both CanESM5 and 2, which is exemplified by their strong linearity. We have expanded our discussion in the results to clarify this point. See L160-164.

**L157: suggest changing "due to increases over the Arctic from sea ice loss" to "due to the sea ice loss over the Arctic"**

Thank you for your suggestion. We have revised this line accordingly. See L175.

**Figure 1: why not also showing LW and SW cloud feedbacks in panel (b)? That will be more direct for readers to see the relative contribution of SW and LW.**

As noted above, Figure 1 now shows the cloud feedback split into its LW and SW components.

**L173: I think it should be "Figure 2" not "Figure 3" in the bracket.**

Thank you for pointing this out. Figure 2, along with this section, has been revised to be consistent with other major structural changes.

**L209: "The situation for the non-low …" – This sentence is not clear to me. Please consider rephrasing.**

Thank you for your suggestion. This sentence has been removed, as the references to the CFMIP2 ensemble results have been removed from Figure 2. Our rationale for removing these results is that we lacked an analogue for CanESM5, and it only served as extra information that did not inform the question of differences between CanESM5 and 2.

**Table 1 caption: should be "amip-future4K" not "amip-future-4K"**

Thank you for pointing this out. Table 1 from the original submission has been removed to maintain consistency with the revised section 3.3.

---

## Author Response (AR2)

**Reviewer 2 Comments and Responses**

**Minor Comment - Figure 5c and d: Why do you choose Years 130-150 to calculate EIS response rather than the full 150 years? Since you are studying the cloud feedback, why not showing the EIS response normalized by the global-mean surface temperature change?**

Thank you for pointing out this inconsistency. We have revised the bottom two panels of Figure 4 as you suggested, where the EIS responses are normalized by the global mean surface temperature response for further parity to cloud feedbacks. We chose to use the latter 20 years of the Abrupt-4xCO$_2$ experiment as the response (relative to the control experiment) as it would bring out the strongest signal given the more positive surface temperature response by the end of the simulation. Using all 150 years could potentially mask this signal given the lack of response in the first few decades relative to the control period.

**Minor comment - I feel the abbreviation 'CFP' is slightly strange because it is never used in the community as far as I know. Why not keep it as the 'cloud feedback parameter' or lambda?**

All instances of the term 'CFP' have been replaced with the full term— net climate feedback parameter.

**Technical Revisions**

All technical revisions you have pointed out have been rectified, thank you for pointing these out.